# Impact of law enforcement and increased traffic fines policy on road traffic fatality, injuries and offenses in Iran: Interrupted time series analysis

**Milad Delavary Foroutaghe[1], Abolfazl Mohammadzadeh Moghaddam[1]\*, Vahid Fakoor[2]**

**1** Department of Civil Engineering, Faculty of Engineering, Ferdowsi University of Mashhad, Mashhad, Razavi Khorasan, Iran, **2** Department of Statistics, Faculty of Mathematical Sciences, Ferdowsi University of Mashhad, Mashhad, Razavi Khorasan, Iran

\* ab–moghadam@um.ac.ir

**Data Availability Statement:** All relevant data are within the paper and its Supporting Information files.

## Abstract

### Background

Road traffic law enforcement was implemented on 1st April 2011 (the first intervention) and traffic ticket fines have been increased on 1st March 2016 (the second intervention) in Iran. The aim of the current study was to evaluate the effects of the law enforcement on reduction in the incidence rate of road traffic fatality (IRRTF), the incidence rate of road traffic injuries (IRRTI) and the incidence rate of rural road traffic offenses (IRRRTO) in Iran.

### Methods

Interrupted time series analysis was conducted to evaluate the impact of law enforcement and increased traffic tickets fines. Monthly data of fatality on urban, rural and local rural roads, injuries with respect to gender and traffic offenses namely speeding, illegal overtaking and tailgating were investigated separately for the period 2009–2016.

### Results

Results showed a reduction in the incidence rate of total road traffic fatality (IRTRTF), the incidence rate of rural road traffic fatality (IRRRTF) and the incidence rate of urban road traffic fatality (IRURTF) by –21.44% (–39.3 to –3.59, 95% CI), –21.25% (–31.32 to –11.88, 95% CI) and –26.75% (–37.49 to –16, 95% CI) through the first intervention which resulted in 0.383, 0.255 and 0.222 decline in casualties per 100 000 population, respectively. Conversely, no reduction was found in the incidence rate of local rural road traffic fatality (IRLRRTF) and the IRRTI. Second intervention was found to only affect the IRURTF with –26.75% (–37.49 to –16, 95% CI) which led to 0.222 casualties per 100 000 population. In addition, a reduction effect was observed on the incidence rate of illegal overtaking (IRIO) and the incidence rate of speeding (IRS) with –42.8% (–57.39 to –28.22, 95% CI) and –10.54% (–21.05 to –0.03, 95% CI which implied a decrease of 415.85 and 1003.8 in monthly traffic offenses per 100 000 vehicles), respectively.

**Funding:** Milad Delavary Foroutaghe, Abolfazl Mohammadzadeh Moghaddam. This work was supported by Ferdowsi University of Mashhad [grant number: 3/45430]. Ferdowsi University of Mashhad -http://um.ac.ir/. This work was supported about 250$ by Ferdowsi University of Mashhad for data collection and specific study [grant number: 3/45430]. The funders had no role in study design, data collection and analysis, decision to publish, or preparation of the manuscript.

**Competing interests:** The authors have declared that no competing interests exist.

## Conclusion

Time series analysis suggests a decline in IRTRTF, IRRRTF, and IRURTF caused by the first intervention. However, the second intervention found to be only effective in IRURTF, IRIO, and IRS with the implication that future initiatives should be focused on modifying the implementation of the traffic interventions.

## 1. Introduction

According to the World health organization (WHO) report in 2018, approximately 3400 road users were killed in traffic accidents on a daily basis in which low and middle-income countries accounted for more than 90% of road traffic fatality [1]. WHO report showed a constant rate of traffic accidents despite population growth throughout the world which proves interventions could be effective in road safety [2]. The report indicated that a 5% reduction in mean speed could lead to 30% decrease in fatal accidents. The use of helmet can reduce the risk of death and severe injury by 40% and 70%, respectively. Moreover, seat belt use reduces the risk of fatal injury by 50% for front seat occupants and by 75% for rear seat occupants [2]. An observational study in Russia on the risk factors under global road safety programme (i.e. seat belt use and speed) led to an increase in seat belt use and a reduction in speeding between October 2010 and March 2013 [3].

Based on Iranian Traffic Police records, every 58 minutes, a road user has been killed in Nowruz 2018 (from 21 March 2018 to 4 April 2018), resulting in 374 traffic casualties [4]. On the first day of April 2011, law enforcement for reducing traffic offenses was implemented. Some of the most important enforcements were related to speeding, illegal overtaking and drink–driving. In addition, since 1$^{st}$ March 2016, traffic ticket fines have been increased. For instance, fine regarding drink–driving has been quadrupled (i.e. from 1 million to 4 million Rials) [5].

Time series analysis can be utilized as a macro study to investigate the policy [6,7,8,9,10,11]. While, temporal and spatio-temporal multivariate random-parameters Tobit models are some of the methods that can be used in the micro level [12,13]. Interrupted time series analysis has been used in various fields bu using autoregressive integrated moving average(ARIMA) methodology which was intoduced by Box–Jenkins in 1976 [14]. For instance, Hansen et al. [15] examined the effect of daylight savings time transition on the incidence rate of unipolar depressive episodes. Hansen et al. [16] studied the effects of Breivik attacks, whose murders accounted for 77 adults and children in Norway, on the rate of trauma- and stressor-related disorders in Denmark. Brals et al. [17] utilized a controlled interrupted time-series to examine the impact of the health insurance and health facility-upgrades on hospital deliveries in rural Nigeria between 1 May 2005 to 30 April 2013.

In addition, Olsen et al. [18] studied the impact of new urban motorway extension on the number of road traffic accidents (RTAs) on local non–motorway roads of Scotland between 1997–2014. Results showed that reduction in RTAs was not associated with the motorway extension. Steinbach et al. [19] found not much evidence of the detrimental effects of dimming, part–night lighting, switch off or changes to white light/LEDs on road accidents/crime in England and Wales. A study by Morrison et al. [20] in the United States revealed that using Uber as an intervention had a reduction effect on alcohol–related accidents in Portland and San Antonio cities. Sebego et al. [21] indicated that the decline in accidents occurred when

educational policies were implemented to reduce alcohol consumption and improve road safety.

Time series analysis, also, has been conducted to evaluate the traffic intervention on fatality and injuries caused by road accidents. For example, Lim and Chi [22] focused on the impact of mobile phone ban in the U.S on reducing fatal crashes involving young drivers aged 14–20. The finding indicated that the ban was only effective in reducing fatal crashes. Lovenheim and Steefel [23] studied the effect of state-level Sunday alcohol sales restrictions on fatal accidents using American time use survey data. The group whose drink driving behaviour was most affected by the laws was underage men and, also, no effect of blue laws on the location of consumption was observed. A study regarding U.S. Child Safety Seat Laws (which have steadily increased mandatory child safety seat restraint were assessed in the United States over the past 35 years) found that the laws saved up to 39 children per year [24]. Lee et al. [25] investigated associations of marijuana law changes and marijuana-involved fatal crashes in the United States in 2018. They noticed no significant changes in the number of marijuana-related crashes after medical legalisation only. Nevertheless, an increased number of marijuana-related crashes were found after the marijuana law changed. The impact of rising gasoline prices which increased new motorcycle sales on fatalities was estimated with ARIMA regression in the United States between 1984–2009. This study introduced evidence that gasoline prices could act as the increasing incentives to purchase motorcycles, leading to a rise in fatalities from motorcycle crashes [26]. Botswana evaluated the effects of traffic policies and alcohol consumption reduction on the decreased incidence rate of traffic fatality and injuries between 2004–2011. Beatriz et al. [27] studied the effect of legal blood alcohol concentration (BAC) reduction in traffic-related fatality and morbidity between January 2003 and December 2014 in Chile and found that alcohol-related injuries were reduced. Furthermore, deregulation policies of the driving licence application process which was proved to facilitate obtaining the licence in Korea had a statistically significant association with the increase in incidence rate of death, injuries, and collisions [28]. Grundy et al. [29] investigated the role of 20 mph traffic speed zones in road traffic injuries between in London, 1986–2006. Results revealed that slower motor vehicle speed records were more successful in reducing the severity of injury rather than frequency of collisions. Traffic interventions can have different outcomes with respect to samples. For instance, Otero et al. [30] evaluated the effect of BAC reduction and increase in driver's licence suspension for traffic offenders on traffic fatality and injuries in Chile, 2009–2014. They found significant reduction in injuries only; thus, unlike prior study, frequency of collisions and injuries has been decreased. Chen et al. [31] used random parameters bivariate ordered probit model to assess potential factors impact the level of injury sustained by two drivers involved in the same rear-end crash between passenger cars. The results showed that driver age, gender, vehicle, airbag /seat belt use and traffic flow were found to impact injury severity for both drivers. Chen et al. [32] studied accident data involving trucks on rural highway to evaluate the difference in driver-injury severity between single- and multi-vehicle accidents by using mixed logit models. It is found that the snow road surface and light traffic indicators will be better modelled as random parameters in SV and MVmodels, respectively.

Previous studies have mainly examined the impact of some interventions, such as seat belt use and alcohol and recreational drugs on road traffic fatality and injuries. Evidence on the impact of law enforcement packages and increasing traffic tickets fines on road traffic fatality, injuries and especially traffic offenses calls for further research. Moreover, most of these studies have been conducted in the developed countries and very few studies have been undertaken in the developing countries. The aim of the current study is to evaluate the impacts of law enforcement and increased traffic tickets fine programmes on fatality in urban, rural and local

rural road networks, on injuries by gender in Iran. In addition, the effect of increased traffic fines on traffic offenses including speeding, illegal overtaking and tailgating investigated in rural areas. To do so, interventional variables such as level shift, delay level shift, additive and temporary change were considered in the time series modelling. In addition, in order to reach the aim of 10 000 casualties by the year 2027, traffic interventions on rural, urban and local rural roads were simulated based on interrupted time series analysis.

## 2. Methods

### 2.1 Data collection

Due to having accessibility to sufficient monthly data for road traffic fatality and injuries for time period March 2009 to February 2016, a time series analysis could have been conducted on data obtained from Iranian Legal Medicine Organization (ILMO). Figs 1 and 2 show time series for IRRTF and IRRTI, respectively. ILMO classifies the fatality based on urban, rural and local rural roads. It also classifies injuries based on gender; however, due to the study limitations, deaths by gender and location of injuries have not been considered for the modelling. In this study, 1[st] of April 2011 and 1[st] of March 2016 representing law enforcement and increased traffic ticket fines, respectively, were considered as interventional points. In Iran, traffic fatality is attributed to those who are killed at the scene or maximally within 30 days after the accident [33].

Each of the points in the time series of IRRTF and IRRTI was calculated per 100 000 population per month. Population per month was obtained through sum total of prior month population and rate of death and birth. National Organization for Civil Registration of Iran publishes data on births and deaths monthly [35].

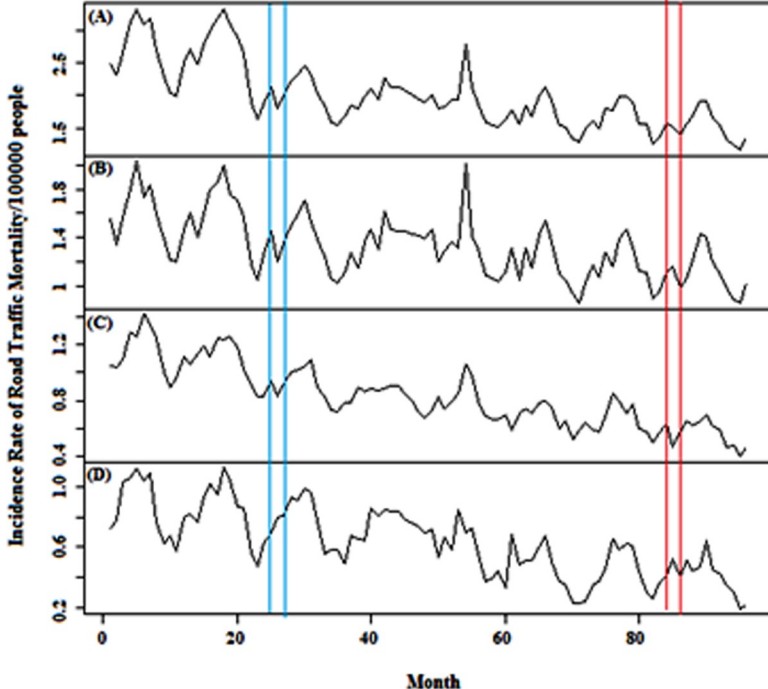

**Fig 1.** Monthly incidence rate of traffic fatality for (A) total, (B) rural, (C) urban and (D) local rural roads in Iran, 2009–2016. Blue and Red vertical lines indicate law enforcement on road traffic on 1 April 2011 and 1 April 2016, respectively.

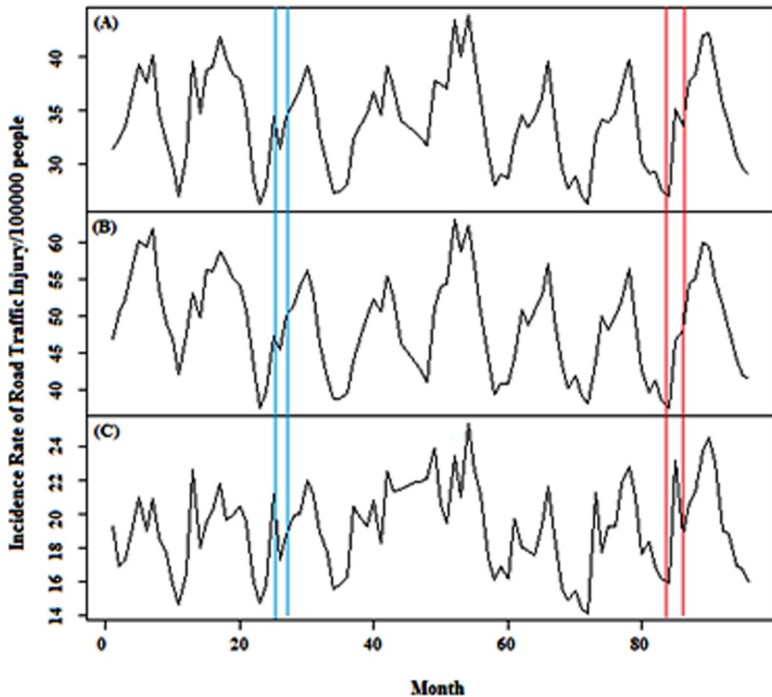

**Fig 2.** Monthly incidence rate of (A) total, (B) male- and (C) female-specific road traffic injuries in Iran, 2009–2016. Blue and Red vertical lines indicate law enforcement on road traffic on 1 April 2011 and 1 April 2016, respectively (this data partly were used by [34]).

In addition, in order to investigate the effect of law enforcement on IRRRTO, traffic offenses data during March 2010 to February 2016 were obtained from Iran Road Maintenance and Transportation Organization (Fig 3) [36]. Data contained illegal overtaking, speeding and tailgating. Due to unavailability of 2009 and 2010 data, only the second intervention was investigated. Every month, these data are published on Excel spreadsheets for all of the camera zones in the rural roads of Iran. In order to acquire the aforementioned traffic parameters, 3 334 000 points were utilized. Moreover, 13 336 000 points were considered for all of them. It should be noted that traffic offense data was provided per 100 000 vehicles per month.

## 2.2 Statistical analysis

In this study, time series which is a special case of panel data were used to model the traffic intervention with statistical tools. In this regard, panel data which are multi-dimensional data involving measurements over time were used as a methodology in a variety of application in traffic safety. For instance, Chen et al. [37] and Chen et al. [38] employed unbalanced panel data to investigate hourly crash frequency on highway segments.

Traffic law enforcement is the example of an intervention which can affect response variables such as a number of casualties or injuries. In 1975, Box and Tiao suggested a method for estimating the effect of interventions in the dynamic regression framework [39]. The interventional analysis is useful when the exact effect of interventions is of interest or the aim of the analysis is to predict the time series by applying the effect of the intervention. In this study, some of the interventional variables like level shift, delay level shift, additive and temporary change are considered in models [40]. Eventually, level shift model with Eq (1) became

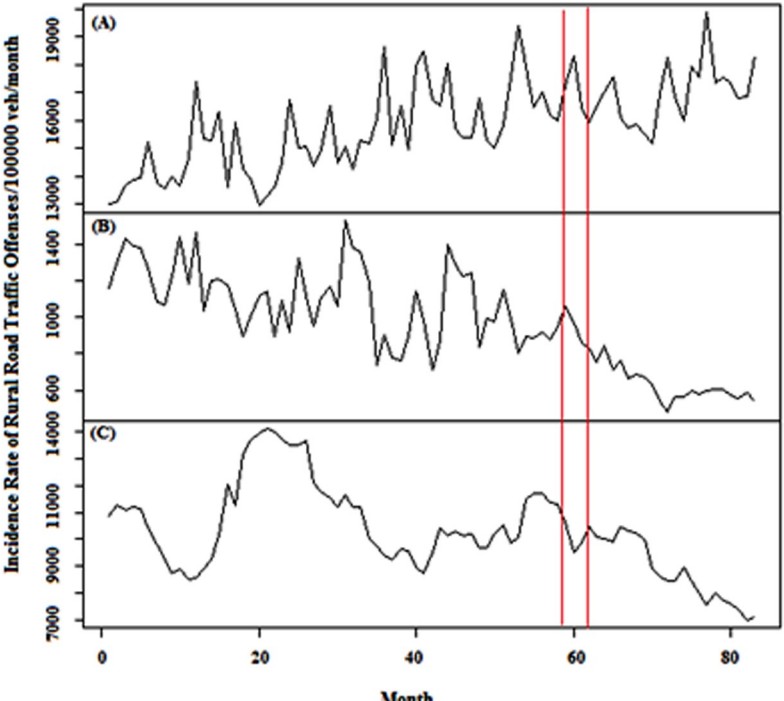

**Fig 3.** Monthly incidence rate of (A) tailgating, (B) illegal overtaking and (C) speeding offenses in rural roads in Iran, 2010–2016. A Red vertical line indicates law enforcement on road traffic on 1 April 2016.

statistically significant.

$$Y_t = a + \omega X_t + N_t \qquad (1)$$

Where $Y_t$, $X_t$, and $N_t$ representing response variable, level shift interventional variable, and a SARIMA model, respectively. If it is assumed that an intervention occurs in 'u' time, a dummy variable can be applied which was equal to 0 before the intervention and become 1 after that [40].

In SARIMA modelling data must be stationary in the mean. In case of non-stationary means, a seasonal difference with a lag of 's' and in case of further requirement, trend difference with a lag of 1 can be employed [41]. As for the monthly data collection in the current study, season considered at lag 12. Stationarity was confirmed with the help of time series plot and Dickey–Fuller (DF) test. This test is one of the unit root tests to check the stationarity of the mean [42,43].

Eventually, for accepting SARIMA model, residuals must follow the white noise. White noise is a discrete signal whose samples are regarded as a sequence of serially uncorrelated random variables with zero mean. Autocorrelation and partial autocorrelation function plots and Ljung–Box test were conducted for discriminating uncorrelated residuals [44]. Furthermore, residual plots were used to determine zero mean [41] and other optional assumptions including the normality of residuals were evaluated with the Kolmogorov-Smirnov (KS) test.

**Ethical approval.** It is not required because this study used a nationally aggregated data derived from the website of Legal Medicine Organization of Iran and the website of National Organization for Civil Registration.

## 3. Results

IRRTF time series of rural, urban and local rural roads and also IRRTI with respect to the gender are shown in Figs 1 and 2, respectively in which blue lines represent first intervention and red lines represent second intervention.

Annual values of IRTRTF between March 2005 to February 2016 have declined from 40.32 to 18.24. This reduction is evident in all types of roads which indicates the positive impact of interventional policies on road safety in Iran. According to the Iranian Traffic Police reports, most of the rural roads' casualties are caused by hazardous moving violations such as speeding and illegal overtaking which their reduction is visible in Fig 3 after March 2016 [4].

The incidence rate of total road traffic injuries (IRTRTI) increased from 398.37 to 426.88 during the years 2005–2016 while the incidence rate of male road traffic injuries (IRMRTI) decreased during the years 2009–2010 and held constant till 2016. Observed IRMRTI values are higher than female (IRFRTI) ones; this has occurred because men use different modes of transport more than women due to their busier life outside the home. The upward trend of IRFRTI during the years 2009–2012 has been balanced due to the law enforcement occurrence since 2013.

Four modes of variables of the interventional time series analysis including level shift, delay level shift, additive and temporary change were evaluated and except for level shift variable, other variables were not described in this study due to their statistically insignificant outcomes. Furthermore, because of uncertainty in the accurate time of dummy variable (i.e. time of enforcing laws) in the model, a time period consisting of three points was used and after evaluating the results of coefficients of the dummy variable, 1$^{st}$ April 2011 and 1$^{st}$ March 2016 were considered for the accurate times of the interventions.

SARIMA models were administered for evaluating the impact of the first and the second interventions on IRRTF, IRRTI and IRRRTO time series. The first intervention had a remarkable reduction effect on IRTRTF, IRRRTF and IRURTF with –21.44% (–39.3 to –3.59, 95% CI), –21.25% (–31.32 to –11.88, 95% CI) and –26.75% (–37.49 to –16, 95% CI) which caused fatality rates reduced by 0.3838, 0.255 and 0.222 casualties per 100 000 population, respectively. IRLRRTF with additive parameter (i.e. 0.1275) experienced a 15.94% increase (–7.31 to 39.19, 95% CI) in the first intervention while the IRURTF evidenced a 26.75% reduction (–37.49 to –16, 95% CI) because of the second intervention (Table 1).

From the Table 2, the impact of the first and the second interventions on IRRTI time series reduction is evident. IRTRTI for the first and the second interventions were estimated around –7.16% (–16.82 to 2.51, 95% CI) and 6.09% (–3.28 to 15.46, 95% CI). The values in the first and the second interventions were –7.88% (–15.84 to 0.08, 95% CI) and 8.07% (–0.24 to 16.39%, 95% CI) and also, –7.49% (–21.92 to 6.95, 95% CI) and 7.38% (–4.32 to 19.07, 95% CI) for men and women, respectively.

IRRRTO time series including tailgating, illegal overtaking and speeding are illustrated in Fig 3. Table 3 shows SARIMA analysis for evaluating the effect of the increased traffic ticket fines on IRRRTO time series. The second intervention led to the reduction in IRIO and IRS by 415.85 and 1003.8 per 100 000 vehicles per month, respectively which is equal to the reduction effect of –42.8% (–57.39 to –28.22, 95% CI) and –10.54% (–21.05 to –0.03, 95% CI). Finally, the reduction in the incidence rate of tailgating (IRT) was –2.57% (–10.26 to 5.12, 95% CI).

Uncorrelated residuals were accepted employing LB test for all of the time series at the 5% significance level (Table 3). Zero mean of the residuals was observed form residual plots to confirm the white noise. Moreover, KS test confirmed the normality of residuals as an optional assumption for all of the time series except IRURTF.

The road safety plan of Safety Commission of the Ministry of Roads and Urban Development in Iran proposed three scenarios in which annual road traffic fatality is expected to reach

**Table 1. The effects of the law enforcement on road traffic fatality, SARIMA models.**

| Output | Monthly average | Estimate | SE | Z | P–value | Change | | LB[c] | KS[d] |
|---|---|---|---|---|---|---|---|---|---|
| | | | | | | Percent Level | 95%CI | | |
| (a)Total | 1593 | | | | | | | | |
| Impact(I1[a]) | | −0.384 | 0.16 | −2.401 | 0.016 | −21.44 | [−39.3, −3.59] | | |
| Noise | | (0,1,3)(1,1,0) | | | | | | 0.75 | 0.331 |
| Impact(I1) | | −0.381 | 0.163 | −2.341 | 0.019 | −21.29 | [−39.48, −3.1] | | |
| Impact(I2[b]) | | −0.096 | 0.174 | −0.555 | 0.579 | −6.38 | [−29.39, 16.62] | | |
| Noise | | (0,1,3)(0,1,1) | | | | | | 0.74 | 0.35 |
| (b) Rural | 1007 | | | | | | | | |
| Impact(I1) | | −0.255 | 0.060 | −4.223 | 2.412e−05 | −21.25 | [−31.32, −11.18] | | |
| Noise | | (0,0,2)(1,1,1) | | | | | | 0.41 | 0.14 |
| Impact(I1) | | −0.247 | 0.056 | −4.396 | 1.101e−05 | −20.54 | [−29.89, −11.19] | | |
| Impact(I2) | | −0.114 | 0.06 | −1.903 | 0.057 | −9.83 | [−20.16, 0.5] | | |
| Noise | | (0,0,2)(1,1,1) | | | | | | 0.38 | 0.14 |
| (c) Urban | 452 | | | | | | | | |
| Impact(I1) | | −0.256 | 0.073 | −3.533 | 0.0004 | −30.89 | [−48.39, −13.4] | | |
| Noise | | (1,0,0)(0,1,1) | | | | | | 0.43 | 0.02 |
| Impact(I1) | | −0.222 | 0.045 | −4.972 | 6.642e−07 | −26.75 | [−37.49, −16] | | |
| Impact(I2) | | −0.106 | 0.048 | −2.210 | 0.027 | −22.58 | [−43, −2.15] | | |
| Noise | | (1,0,0)(1,1,0) | | | | | | 0.62 | 0.02 |
| (d) Local rural | 134 | | | | | | | | |
| Impact(I1) | | 0.128 | 0.093 | 1.371 | 0.170 | 15.94 | [−7.31, 39.19] | | |
| Noise | | (1,1,0)(1,0,1) | | | | | | 0.41 | 0.08 |
| Impact(I2) | | −0.072 | 0.093 | −0.778 | 0.436 | −13.87 | [−49.52, 21.79] | | |
| Noise | | (2,0,0)(0,1,1) | | | | | | 0.44 | 0.12 |

[a]Intervention1: 1 April 2011

[b]Intervention2: 1 March 2016

[c]Ljung–Box

[d]Kolmogorov–Smirnov

12 000, 10 000 and 8 000 causalities until 2027 [45]. Considering the current situation, in order to reach this milestone, interventions should be considered. In this study, simulation for interventions in 1st March 2019, 1st March 2022 and 1st March 2025 was considered with the level shift effects of 0.15, 0.15 and 0.1 for IRRRTF, IRURTF and IRLRRTF, respectively. The reason behind choosing these values was the results of the analysis of traffic safety interventions in this study. Selection of 1st March was for the implementation of the intervention and three years interval for each of interventions was chosen in order to reach at least one of the mentioned scenarios until 2027 and also because of obtained experiences in the current study.

Eventually, results of this simulation are shown in Table 4 representing the monthly incidence rate of road traffic fatality in urban, rural and local rural roads. IRRRTF, IRURTF, and IRLRRTF annual values will reach 7,809, 966 and 389, respectively till 2027. Therefore, in case of applying the three aforementioned interventions, total fatality would be 9,146 road users and shows that the first two scenarios will be approachable.

## 4. Discussion and conclusion

Law enforcement on 1st April 2011 and increased traffic ticket fines on 1st March 2016 have been implemented as interventions to reduce the incidence rate of road traffic fatality and

**Table 2. The effects of the law enforcement on road traffic injuries, SARIMA models.**

| Output | Monthly average | Estimate | SE | Z | P–value | Change | | LB[c] | KS[d] |
|---|---|---|---|---|---|---|---|---|---|
| | | | | | | Percent Level | 95%CI | | |
| (a)Total | 26037 | | | | | | | | |
| Impact(I1[a]) | | −2.253 | 1.521 | −1.481 | 0.139 | −7.16 | [−16.82, 2.51] | | |
| Noise | | (0,1,1)(0,1,1) | | | | | | 0.43 | 0.8 |
| Impact(I2[b]) | | 2.141 | 1.647 | 1.300 | 0.194 | 6.09 | [−3.28, 15.46] | | |
| Noise | | (0,1,1)(0,1,1) | | | | | | 0.43 | 0.76 |
| (b) Male | 18827 | | | | | | | | |
| Impact(I1) | | −3.245 | 1.642 | −1.976 | 0.048 | −7.13 | [−14.35, 0.09] | | |
| Noise | | (2,0,0)(0,1,1) | | | | | | 0.10 | 0.76 |
| Impact(I1) | | −3.585 | 1.811 | −1.98 | 0.048 | −7.88 | [−15.84, 0.08] | | |
| Impact(I2) | | 3.778 | 1.946 | 1.942 | 0.052 | 8.07 | [−0.24, 16.39] | | |
| Noise | | (2,0,0)(1,1,0) | | | | | | 0.11 | 0.87 |
| (c) Female | 7210 | | | | | | | | |
| Impact(I1) | | −1.289 | 1.242 | −1.038 | 0.3 | −7.49 | [−21.92, 6.95] | | |
| Noise | | (1,1,1)(0,0,1) | | | | | | 0.15 | 0.93 |
| Impact(I2) | | 1.707 | 1.353 | 1.2614 | 0.207 | 7.38 | [−4.32, 19.07] | | |
| Noise | | (1,1,1)(0,0,1) | | | | | | 0.21 | 0.87 |

[a]Intervention1: 1 April 2011

[b]Intervention2: 1 March 2016

[c]Ljung–Box

[d]Kolmogorov–Smirnov

**Table 3. The effects of the law enforcement on road traffic offenses, SARIMA models.**

| Output | Monthly average | Estimate | SE | Z | P–value | Change | | LB[c] | KS[d] |
|---|---|---|---|---|---|---|---|---|---|
| | | | | | | Percent Level | 95%CI | | |
| (a) Tailgating | 15870 | | | | | | | | |
| Impact(I2[a]) | | −469.797 | 703.728 | −0.668 | 0.504 | −2.57 | [−10.26, 5.12] | | |
| Noise | | (1,1,0)(1,1,1) | | | | | | 0.59 | 0.87 |
| (b) Illegal Overtaking | 973 | | | | | | | | |
| Impact(I2) | | −415.85 | 70.859 | −5.869 | 4.393e−09 | −42.8 | [−57.39, −28.22] | | |
| Noise | | (1,0,0)(1,0,0) | | | | | | 0.86 | 0.96 |
| (c) Speeding | 10311 | | | | | | | | |
| Impact(I2) | | −1.004e+03 | 5.004e+02 | −2.006 | 0.044 | −10.54 | [−21.05, −0.03] | | |
| Noise | | (1,0,0)(0,0,1) | | | | | | 0.68 | 0.99 |

[a]Intervention1: 1 April 2011

[b]Intervention2: 1 March 2016

[c]Ljung–Box

[d]Kolmogorov–Smirnov

**Table 4. Predicted monthly incidence rate of road traffic fatality for 2027.**

| Road / Month | Rural | Urban | Local rural | Total |
|---|---|---|---|---|
| March | 665 | 38 | 34 | 737 |
| April | 506 | 53 | 24 | 583 |
| May | 664 | 127 | 40 | 831 |
| June | 756 | 154 | 49 | 959 |
| July | 877 | 158 | 55 | 1090 |
| August | 955 | 162 | 64 | 1181 |
| September | 800 | 138 | 53 | 991 |
| October | 669 | 81 | 32 | 782 |
| November | 561 | 11 | 16 | 588 |
| December | 422 | 1 | 8 | 431 |
| January | 398 | 15 | 5 | 418 |
| February | 536 | 28 | 9 | 573 |
| Total | 7809 | 966 | 389 | 9164 |

injuries in Iran. The aim of the study was to examine the effects of these interventions on road traffic fatality, injuries, and offenses. In addition, in order to reach 10 000 casualties by the year 2027 according to the plan of Iranian committee on road safety, a simulation was conducted based on the interrupted time series analysis with level shift interventions on urban, rural and local rural roads [45]. Traffic interventions were found to be effective in decreasing traffic accidents. The results of interrupted time series analysis revealed a reduction of total road traffic fatality in Iran, especially on urban and rural roads. Urban roads witnessed the highest reduction. Quite the opposite, no reduction effect was observed on the local rural roads.

In the first interventional analysis, despite having no significant reduction in total injuries, a higher reduction effect has been observed in injured men probably because they have higher exposure to traffic laws as they use motor vehicles more often and have a higher outdoor activity than women. Regarding the failure of the intervention to reduce injuries, one can assume that safety policies have been effective only on the severity of injury and therefore it only reduced the associated fatality.

The second intervention was found effective in IRURTF. On rural roads, however, no obvious reduction was observed probably due to the ineffectiveness of the intervention on some of the traffic offenses like tailgating. In addition, this intervention did not influence the injuries. The results confirmed the higher impact of the first intervention. The second intervention could have been more effective if social or educational advertisements had been used instead.

Based on the interrupted time series simulation, in case of applying level shift intervention with 0.15, 0.15 and 0.1 for rural, urban and local rural roads on 1st March 2019, 1st March 2022 and 1st March 2025, respectively the aim of having no more than 10 000 deaths until the year 2027 would be approachable.

This study was unable to account for definite effects of the interventions on the frequency of collisions due to lack of accessibility to the number of collisions resulting in death or injury. ILMO does not publish deaths based on gender, and injuries based on the road type. Therefore, it is worthwhile to inspect the effect of interventions on fatality and injuries with respect to gender and road type in the future studies. Moreover, RMTO has provided rural road traffic offenses data since 2010 and thus it was unfeasible to study the effect of the first intervention on these data.

In conclusion, as previous studies did not evaluate the effects of implementation of traffic safety interventions in Iran, this study highlights the effects of law enforcement and increased traffic fines on traffic fatality, injuries, and offenses. Law enforcement was only successful in reducing fatality while there was no obvious change in the total number of injuries. Furthermore, increased traffic fines as another intervention was also unable to achieve its target of reducing associated deaths and injuries and this happened probably due to the improper implementation of this policy to reduce hazardous moving violations.

## Supporting information

**S1 File. Total aggregated data–mortalities.**
(RAR)

**S2 File. Total aggregated data–injuries.**
(RAR)

**S3 File. Total aggregated data–traffic offenses.**
(RAR)

## Author Contributions

**Conceptualization:** Milad Delavary Foroutaghe, Abolfazl Mohammadzadeh Moghaddam.

**Data curation:** Milad Delavary Foroutaghe, Abolfazl Mohammadzadeh Moghaddam.

**Formal analysis:** Milad Delavary Foroutaghe.

**Funding acquisition:** Milad Delavary Foroutaghe, Abolfazl Mohammadzadeh Moghaddam.

**Investigation:** Vahid Fakoor.

**Methodology:** Milad Delavary Foroutaghe, Abolfazl Mohammadzadeh Moghaddam, Vahid Fakoor.

**Project administration:** Abolfazl Mohammadzadeh Moghaddam.

**Software:** Milad Delavary Foroutaghe, Vahid Fakoor.

**Supervision:** Abolfazl Mohammadzadeh Moghaddam.

**Writing – original draft:** Milad Delavary Foroutaghe, Abolfazl Mohammadzadeh Moghaddam.

**Writing – review & editing:** Abolfazl Mohammadzadeh Moghaddam.

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
