## [Decision Letter · Decision Letter 0]

24 Dec 2019

PONE-D-19-32920

Impact of law enforcement and increased traffic ticket fines policy on road traffic mortality, injuries and offenses in Iran: Interrupted time series analysis

PLOS ONE

Dear Dr. Mohammadzadeh Moghaddam,

Thank you for submitting your manuscript to PLOS ONE. After careful consideration, we feel that it has merit but does not fully meet PLOS ONE’s publication criteria as it currently stands. Therefore, we invite you to submit a revised version of the manuscript that addresses the points raised during the review process.

We would appreciate receiving your revised manuscript by Feb 07 2020 11:59PM. To enhance the reproducibility of your results, we recommend that if applicable you deposit your laboratory protocols in protocols.io, where a protocol can be assigned its own identifier (DOI) such that it can be cited independently in the future. For instructions see: http://journals.plos.org/plosone/s/submission-guidelines#loc-laboratory-protocols

We look forward to receiving your revised manuscript.

Kind regards,

Feng Chen

Academic Editor

PLOS ONE

Journal Requirements:

4. Your ethics statement must appear in the Methods section of your manuscript. If your ethics statement is written in any section besides the Methods, please move it to the Methods section and delete it from any other section. Please also ensure that your ethics statement is included in your manuscript, as the ethics section of your online submission will not be published alongside your manuscript.

5. Please remove citations from the abstract. Please also make sure that all submission guidelines have been followed: " ext-link-type="uri" xlink:type="simple">https://journals.plos.org/plosone/s/submission-guidelines"

6.  We note that you have indicated that data from this study are available upon request. PLOS only allows data to be available upon request if there are legal or ethical restrictions on sharing data publicly. For more information on unacceptable data access restrictions, please see http://journals.plos.org/plosone/s/data-availability#loc-unacceptable-data-access-restrictions.

7. We noted in your submission details that a portion of your manuscript may have been presented or published elsewhere: Very small Part of data, just injuries data, was published for forecasting, not intervention analysis, in Plose One:with the following doi:

https://doi.org/10.1371/journal.

pone.0216462

Reviewers' comments:

Reviewer's Responses to Questions

**Comments to the Author**

1. Is the manuscript technically sound, and do the data support the conclusions?

Reviewer #1: Partly

Reviewer #2: Partly

2. Has the statistical analysis been performed appropriately and rigorously? 

Reviewer #1: No

Reviewer #2: N/A

3. Have the authors made all data underlying the findings in their manuscript fully available?

Reviewer #1: Yes

Reviewer #2: Yes

4. Is the manuscript presented in an intelligible fashion and written in standard English?

Reviewer #1: Yes

Reviewer #2: Yes

5. Review Comments to the Author

Reviewer #1: This study conducted a time series analysis of the impacts of law enforcement and increased traffic ticket fines policy on road traffic mortality, injuries and offenses in Iran. The research topic is worth of investigation. However, several revisions may be required before its publication.

First, the Introduction section should be re-structured. For instants, some alcohol-related studies reviewed in the section are not related to road traffic safety. Thus, they should be removed from the section.

In the proposed statistical model, accommodating temporal correlations is reasonable. However, the linear model may not be appropriate. Using Tobit-based approaches for modeling crash rates may be more suitable, such as the following works:

Jointly modeling area-level crash rates by severity: A Bayesian multivariate random-parameters spatio-temporal Tobit regression. Transportmetrica A: Transport Science, 15 (2): 1867-1884.

Incorporating temporal correlation into a multivariate random parameters Tobit model for modeling crash rate by injury severity. Transportmetrica A: Transport Science, 14 (3): 177-191.

A Bayesian spatial random parameters Tobit model for analyzing crash rates on roadway segments. Accident Analysis Prevention, 100, 37-43.

As mentioned in the above papers, in addition to temporal correlations, spatial correlations can also be considered in the models.

Reviewer #2: The topic of this paper is important. The results are meaningful and useful. There are several suggestions to improve this paper.

1. “mortality” is typically replaced by “fatality” in this field.

2. What’s the tendency of the population and traffic volume in Iran in the time period?

3. Line 54-64, the information from World health organization (2015) report is too lengthy. And the reference (World Health Organization, 2015) need to be mentioned on line 54.

4. For the influence of law enforcement on injury severity, more references are needed. For example, the following ones.

[1] Investigation on the Injury Severity of Drivers in Rear-End Collisions Between Cars Using a Random Parameters Bivariate Ordered Probit Model, International Journal of Environmental Research and Public Health, 2019, 16(14) , 2632.

[2] “Injury severities of truck drivers in single- and multi-vehicle accidents on rural highway”, Accident Analysis and Prevention, 2011, 43(5), 1677-1688.

5. “Also, in order to reach the aim of 10 000 casualties by the year 2027” There need to be a reference and the nation should be mentioned.

6. For the methodology, the authors need to at least mention the panel-data models which combine time-serial and cross-section models. The following are some references.

[3] Analysis of hourly crash likelihood using unbalanced panel data mixed logit model and real-time driving environmental big data. 2018, JOURNAL OF SAFETY RESEARCH. 65: 153-159.

[4] “Investigating the Differences of Single- and Multi-vehicle Accident Probability Using Mixed Logit Model", Journal of Advanced Transportation, 2018, UNSP 2702360.

[5] “Crash Frequency Modeling Using Real-Time Environmental and Traffic Data and Unbalanced Panel Data Models”, International Journal of Environmental Research and Public Health, 2016, 13(6), 609.

6. PLOS authors have the option to publish the peer review history of their article (what does this mean?). If published, this will include your full peer review and any attached files.

Reviewer #1: No

Reviewer #2: No

---

## [Author Response · Author response to Decision Letter 0]

2 Mar 2020

Dear Professor Feng Chen, Ph.D

Associate Editor

Plos One

OBJECT: Resubmission of manuscript PONE-D-18-20018

Re: Impact of law enforcement and increased traffic ticket fines policy on road traffic mortality, injuries and offenses in Iran: Interrupted time series analysis 

Dear Editor,

We would like to thank you for attention to our manuscript. Also, we would like to thank the Associate Editor and the Reviewers for their careful reading of our manuscript and their insightful comments. In view of these, we have revised our manuscript. It is promising to realize that the reviewers have suggested a number of promoting modifications. The comments have been noted and we have tried to revise the paper accordingly. Below is a point-by-point response to each issue and concern raised by the Associate Editor and both Reviewers (original comments from reviewers appear in red color, the responses in black). It should be noted that the modifications in the manuscript are highlighted with red color tracking version of word 2013. In addition, a final version of the manuscript including all type of corrections is attached. 

We hope that the revised version of the manuscript which embraces all the constructive comments kindly raised by reviewer meet your positive view. 

Please do not hesitate to contact me if you think that the manuscript needs further modifications or clarifications.

Yours sincerely,

Abolfazl Mohammadzadeh Moghaddam 

Tel:

Office: 0985138805026

Mobile: 0989158969602

E-mail: ab-moghadam@um.ac.ir/Mohammadzadeh.abolfazl@gmail.com

Associate Editor:

We would like to thank you for your comments. Below please see our answers in which we have revised our manuscript.

Associate Editor’s Comments to the Author

Point-by-point responses to the issues raised by the Associate Editor:

1- When submitting your revision, we need you to address these additional requirements.

ANSWER: As you truly said, according to affiliation formatting guidelines, numbers was used instead of letters in lines 4-13. In addition, the contact of corresponding author is in line 14.

2- Please include captions for your Supporting Information files at the end of your manuscript, and update any in-text citations to match accordingly. Please see our Supporting Information guidelines for more information: http://journals.plos.org/plosone/s/supporting-information.

ANSWER: Supporting information were added in lines 406-410 with the revised names. This information is S1 file, S2 file and S3 file with the names of Total aggregated data-Mortalities, injuries and traffic offenses respectively. It should be mentioned that these files are in ZIP version.

3- We suggest you thoroughly copyedit your manuscript for language usage, spelling, and grammar. If you do not know anyone who can help you do this, you may wish to consider employing a professional scientific editing service. 

ANSWER: Thanks for your suggestion. Although the reviewers said there is no problems with the language, spelling ang grammar of paper, the authors themselves and with the help of a proofreader reviewed the paper again and correct some mistakes. 

4- Your ethics statement must appear in the Methods section of your manuscript. If your ethics statement is written in any section besides the Methods, please move it to the Methods section and delete it from any other section. Please also ensure that your ethics statement is included in your manuscript, as the ethics section of your online submission will not be published alongside your manuscript.

ANSWER: Thank you for your comment. The authors statement including funding, ethical approval and etc. were moved bellow method section. But, as authors said in the ethic statement, this is not required because this study used a nationally aggregated data derived from the website of Legal Medicine Organization of Iran and the website of National Organization for Civil Registration.

5- Please remove citations from the abstract. Please also make sure that all submission guidelines have been followed: https://journals.plos.org/plosone/s/submission-guidelines"

ANSWER: The footnotes for law enforcement and increasing of traffic ticket fines were removed and instead of that, we used these footnotes in the context in line 29-30.

6- We note that you have indicated that data from this study are available upon request. PLOS only allows data to be available upon request if there are legal or ethical restrictions on sharing data publicly. For more information on unacceptable data access restrictions, please see http://journals.plos.org/plosone/s/data-availability#loc-unacceptable-data-access-restrictions.

ANSWER: As the authors said in ethic statement, nationally aggregated data derived from the website of Legal Medicine Organization of Iran and the website of National Organization for Civil Registration. However, aggregated and final data were uploaded and supporting information section was added in line 406-410. S1 file, S2 file and S3 file as data were displayed in this section.

7- We noted in your submission details that a portion of your manuscript may have been presented or published elsewhere: Very small Part of data, just injuries data, was published for forecasting, not intervention analysis, in Plose One:with the following doi:

https://doi.org/10.1371/journal.

pone.0216462

ANSWER: Thanks for your comment and concern about avoiding dual publication. As you said, the paper was formally published with peer-review. However, data used in current paper include time series of mortality, injury and traffic offenses in Iran. In our previous paper with the following doi:https://doi.org/10.1371/journal.pone.0216462, authors used just injury data for forecasting time series with SARIMA models. However, in this study, we used dynamic regression for evaluating, not forecasting, the impact of traffic interventions including law enforcement and increasing traffic ticket fines with mortality, injury and traffic offenses of roads in Iran. So, not only the scope of this paper is different, but also the methodology is different, too. In addition, the data in this study is more than previous study (in current study we evaluate more than 10 time series but in previous paper we just forecast three time series). However, in cover letter, authors mentioned why current study is not dual publication with some logical reasons and also we added the above published paper as a reference for the fig 2 with “this data partly were used by Delavar et al. (2019))”.

Reviewer #1:

Review’s Comments to the Author

Point-by-point responses to the issues raised by the Reviewer #1:

This study conducted a time series analysis of the impacts of law enforcement and increased traffic ticket fines policy on road traffic mortality, injuries and offenses in Iran. The research topic is worth of investigation. However, several revisions may be required before its publication.

We appreciate the reviewer's valuable comments. We have tried to enhance the manuscript by addressing your comments. In the following section our responses are presented:

The detailed comments are as follows:

1- The Introduction section should be re-structured. For instants, some alcohol-related studies reviewed in the section are not related to road traffic safety. Thus, they should be removed from the section.

ANSWER: We are in agreement with the reviewer comment. So, those researches related to alcohol effect were removed from introduction section. Also, the introduction was reorganized which is obvious in track-change version. In addition, lines 348-350, 353-363 and 369-377 were moved to introduction section which is in the paragraph about traffic interventions and their impact on injuries and mortalities. In addition, some relevant references related to evaluating the impact of law enforcement on road traffic injuries were added in lines 144-162. Also, studies suggested by the reviewer were helpful for enriching the introduction and were written in lines 163-170. 

“Botswana evaluated effects of traffic policies and alcohol consumption reduction on the decreased incidence rate of traffic fatality and injuries between 2004-2011. Beatriz et al. (2017) studied the effect of legal blood alcohol concentration (BAC) reduction in traffic-related fatality and morbidity between January 2003 and December 2014 in Chile and found that alcohol-related injuries were reduced. In addition, deregulation policies of the driving license application process which was proved to facilitate obtaining the license in Korea had a statistically significant association with the increase in incidence rate of death, injuries, and collisions (Oh et al., 2016). Grundy et al. (2015) investigated the role of 20 mph traffic speed zones in road traffic injuries between 1986-2006 in London. Results revealed that slower motor vehicle speeds were more successful in reducing the severity of injury rather than frequency of collisions. Traffic interventions can have different outcomes with respect to samples. For instance, Otero et al. (2017) evaluated the effect of BAC reduction and increase in driver's license suspension for traffic offenders on traffic fatality and injuries between 2009-2014 in Chile. They found significant reduction only in injuries; thus, unlike prior study, frequency of collisions and injuries has been decreased.

Chen et al. (2019) used random parameters bivariate ordered probit model to to assess potential factors aecting the level of injury sustained by two drivers involved in the same rear-end crash between passenger cars. The results showed that driver age, gender, vehicle, airbag or seat belt use, traffic flow are found to impact injury severity for both the two drivers. 

Chen et al. (2011) studied accident data involving trucks on rural highway to evaluate the difference in driver-injury severity between single- and multi-vehicle accidents by using mixed logit models. It is found that the snow road surface and light traffic indicators will be better modeled as random parameters in SV and MVmodels respectively. "

2- In the proposed statistical model, accommodating temporal correlations is reasonable. However, the linear model may not be appropriate. Using Tobit-based approaches for modeling crash rates may be more suitable, such as the following works:

Jointly modeling area-level crash rates by severity: A Bayesian multivariate random-parameters spatio-temporal Tobit regression. Transportmetrica A: Transport Science, 15 (2): 1867-1884.

Incorporating temporal correlation into a multivariate random parameters Tobit model for modeling crash rate by injury severity. Transportmetrica A: Transport Science, 14 (3): 177-191.

A Bayesian spatial random parameters Tobit model for analyzing crash rates on roadway segments. Accident Analysis Prevention, 100, 37-43.

As mentioned in the above papers, in addition to temporal correlations, spatial correlations can also be considered in the models.

ANSWER: Thank you for the references and authors reviewed them. 

First, we think that the tobit model or censored regression model is designed to estimate linear relationships between variables when there is either left- or right-censoring in the dependent variable. In this study, univariate models were used and there is no censored data because these data were obtained from Iranian Legal Medicine and Road Ministry from all highways in Iran and include mortality, injury and traffic offenses which were recorded from cameras or collected by Iranian Legal Medicine. 

Secondly, this study used interrupted time series to investigate the changes during studied time period and this methodology was used with researchers across the world to evaluate the intervention base on time domain (Gustafsson et al, 2011; Herttua et al, 2009; Pun et al, 2013; Kisely et al, 2016; Sanchez et al, 2011; Herttua et al,2015). In addition, If the data used in this study were in segments and changed during the time, so authors could consider spatial correlation in the model. But, this study is a macro study with big data to investigate policy of Iranian government in transportation and it is not related to a specific area with a lot of segments. 

Thirdly, considering both time and location domain is like using panel data which are multi-dimensional data involving measurements, observations of multiple phenomena over time. In our study time series of road accident mortality, injury and traffic offenses of Iran which is a special case of panel data were studied. So, other factors which affect mortality or injury for the studied time period are not available to use them as paned data with multi-dimensional.

In addition, as authors said, there are a lot of papers with using interrupted time series like the methodology of this paper such as following articles. It should be noted that in this study, interrupted time series method was extended with using a variety of interventions like level shift and delay level shift. Also, change point detection was added as the process of modeling.

Finally, the above references, that the reviewer suggest, were used in lines 105-108 for more clarification of the differences between methods that can be utilized as macro and micro level.

“Time series analysis can be utilized as macro study to investigate the policy (Gustafsson et al, 2011; Herttua et al, 2009; Pun et al, 2013; Kisely et al, 2016; Sanchez et al, 2011; Herttua et al,2015). While, temporal and spatio-temporal multivariate random-parameters tobit model are some of the methods that can be used as micro level (Zeng et al, 2017; Zeng et al, 2019).”

Zeng, Q., Wen, H., Xin Pei, H. H., Wong, S. C., 2017. Incorporating temporal correlation into a multivariate random parameters Tobit model for modeling crash rate by injury severity, Transportmetrica A: Transport Science.

Zeng, Q., Guo, Q., Wong, S. C., Wen, H., Huang, H., Pei, X., 2019, Jointly modeling area-level crash rates by severity: A Bayesian multivariate randomparameters spatio-temporal Tobit regression, Transportmetrica A: Transport Science.

Gustafsson, N.J., Ramstedt, M.R., 2011. Changes in alcohol-related harm in Sweden after increasing alcohol import quotas and a Danish tax decrease—an interrupted time-series analysis for 2000–2007. International Journal of Epidemiology 40, 432-440.

Herttua, K., Makela, P., Martikainen, P., 2009. An evaluation of the impact of a large reduction in alcohol prices on alcohol-related and all-cause mortality: time series analysis of a population-based natural experiment. International Journal of Epidemiology 40, 441-454.

Pun, V.C., Lin, H., Kim, J.H., et al, 2013. Impacts of alcohol duty reductions on cardiovascular mortality among elderly Chinese: a 10-year time series analysis. Journal of Epidemiology and Community Health 67, 514-8.

Kisely, S., Lawrence, D., 2016. A time series analysis of alcohol-related presentations to emergency departments in Queensland following the increase in alcopops tax. journal of Epidemiology and Community Health 70, 181-186.

Sanchez, A.I., Villaveces, A., Krafty, R.T., Park, T., Weiss, H.B., Fabio, A., Puyana, J.C., et al., 2011. Policies for alcohol restriction and their association with interpersonal violence: a time-series analysis of homicides in Cali, Colombia. International Journal of Epidemiology 40, 1037-1046.

Herttua, K., Makela, P., Martikainen, P., 2015. Minimum Prices for Alcohol and Educational Disparities in Alcohol-related mortality. Epidemiology 26, 337-343.

Reviewer #2:

Review’s Comments to the Author

Point-by-point responses to the issues raised by the the Reviewer #2:

The topic of this paper is important. The results are meaningful and useful. There are several suggestions to improve this paper.

We appreciate the reviewer for his/her positive comments about the subject of our research and your insightful feedbacks about the material. We tried to use this opportunity to increase the quality of the article using the comments of the reviewer.

1- “mortality” is typically replaced by “fatality” in this field.

ANSWER: Thanks for your helpful advice to replace fatality instead of mortality. Some authors in epidemiology studies used mortality instead of fatality which is mentioned bellow. So, authors decided to use this world. But, due to helpful comment of respective reviewer, fatality was used in the paper to be more precise.

Herttua, K., Makela, P., Martikainen, P., 2009. An evaluation of the impact of a large reduction in alcohol prices on alcohol-related and all-cause mortality: time series analysis of a population-based natural experiment. International Journal of Epidemiology 40, 441-454.

Herttua, K., Makela, P., Martikainen, P., 2015. Minimum Prices for Alcohol and Educational Disparities in Alcohol-related mortality. Epidemiology 26, 337-343.

Pun, V.C., Lin, H., Kim, J.H., et al, 2013. Impacts of alcohol duty reductions on cardiovascular mortality among elderly Chinese: a 10-year time series analysis. Journal of Epidemiology and Community Health 67, 514-8.

2- What’s the tendency of the population and traffic volume in Iran in the time period?

ANSWER: National Organization for Civil Registration of Iran publishes data on births and deaths monthly (National Organization for Civil Registration, 2017). According to these data, the trend of Iranian population is positive. In this case, 950,000 people were added to the population each year. Also, the tendency of volume per camera (due to increasing the cameras in Iran, so the trend of aggregated volume is not normalized) is increasing. 

National Organization for Civil Registration(NOCR), 2017. Iran. Available: https://www.sabteahval.ir/en [Accessed].

3- Line 54-64, the information from World health organization (2015) report is too lengthy. And the reference (World Health Organization, 2015) need to be mentioned on line 54.

ANSWER: The manuscript has been reviewed by authors and revised. So, the line 58 and lines 64-67 were removed and the last sentence which is in line 76-79 is moved as the first sentence in the paragraph in lines 59-62. Also, the reference (Word Health Organization) was added in line 65.

4- For the influence of law enforcement on injury severity, more references are needed. For example, the following ones.

[1] Investigation on the Injury Severity of Drivers in Rear-End Collisions Between Cars Using a Random Parameters Bivariate Ordered Probit Model, International Journal of Environmental Research and Public Health, 2019, 16(14) , 2632.

[2] “Injury severities of truck drivers in single- and multi-vehicle accidents on rural highway”, Accident Analysis and Prevention, 2011, 43(5), 1677-1688.

ANSWER: Thanks for your constructive suggestion. Some useful references related to evaluating the impact of law enforcement on road traffic injuries were added in lines 144-162. In addition, papers which the reviewer mentioned are helpful for enriching the introduction and were written in lines 163-170. 

“Botswana evaluated effects of traffic policies and alcohol consumption reduction on the decreased incidence rate of traffic fatality and injuries between 2004-2011. Beatriz et al. (2017) studied the effect of legal blood alcohol concentration (BAC) reduction in traffic-related fatality and morbidity between January 2003 and December 2014 in Chile and found that alcohol-related injuries were reduced. In addition, deregulation policies of the driving license application process which was proved to facilitate obtaining the license in Korea had a statistically significant association with the increase in incidence rate of death, injuries, and collisions (Oh et al., 2016). Grundy et al. (2015) investigated the role of 20 mph traffic speed zones in road traffic injuries between 1986-2006 in London. Results revealed that slower motor vehicle speeds were more successful in reducing the severity of injury rather than frequency of collisions. Traffic interventions can have different outcomes with respect to samples. For instance, Otero et al. (2017) evaluated the effect of BAC reduction and increase in driver's license suspension for traffic offenders on traffic fatality and injuries between 2009-2014 in Chile. They found significant reduction only in injuries; thus, unlike prior study, frequency of collisions and injuries has been decreased.

Chen et al. (2019) used random parameters bivariate ordered probit model to assess potential factors affecting the level of injury sustained by two drivers involved in the same rear-end crash between passenger cars. The results showed that driver age, gender, vehicle, airbag or seat belt use, traffic flow are found to impact injury severity for both the two drivers. 

Chen et al. (2011) studied accident data involving trucks on rural highway to evaluate the difference in driver-injury severity between single- and multi-vehicle accidents by using mixed logit models. It is found that the snow road surface and light traffic indicators will be better modeled as random parameters in SV and MVmodels respectively. "

5- Also, in order to reach the aim of 10 000 casualties by the year 2027” There need to be a reference and the nation should be mentioned.

ANSWER: The reference and country for the sentence “in order to reach the aim of 10000 casualties by the year 2027 are mentioned in lines 322-325. In these lines, it mentioned that the road safety plan of Safety Commission of the Ministry of Roads and Urban Development in Iran proposed 3 scenarios in which annual road traffic fatality is expected to reach 12 000, 10 000 and 8 000 causalities until 2027 (Iranian Committee on Roads Safety, 2018). However, the reference and the nation of above sentence were added and revised sentence is in line 344 and 347.

6- For the methodology, the authors need to at least mention the panel-data models which combine time-serial and cross-section models. The following are some references.

[3] Analysis of hourly crash likelihood using unbalanced panel data mixed logit model and real-time driving environmental big data. 2018, JOURNAL OF SAFETY RESEARCH. 65: 153-159.

[4] “Investigating the Differences of Single- and Multi-vehicle Accident Probability Using Mixed Logit Model", Journal of Advanced Transportation, 2018, UNSP 2702360.

[5] “Crash Frequency Modeling Using Real-Time Environmental and Traffic Data and Unbalanced Panel Data Models”, International Journal of Environmental Research and Public Health, 2016, 13(6), 609.

ANSWER: First, thank you for your constructive suggestion. As you know, panel data are multi-dimensional data involving measurements, observations of multiple phenomena over time. In our study time series of road accident mortality, injury and traffic offenses of Iran were studied. So, other factors which affect mortality or injury for the studied time period are not available to use them as paned data with multi-dimensional. However, time series data can be studied and modeled as special cases of panel data that are in one dimension only. So, above references are useful to enrich and strength the introduction and used in line 226-230. 

“In this study, Time series which is a special case of panel data were used to model the traffic intervention with statistical tools. In this regard, Panel data which are multi-dimensional data involving measurements over time were used as a methodology in a variety of application in traffic safety. For instance, Chen et al. (2016) and Chen et al. (2018) used unbalanced panel data to investigate hourly crash frequency on highway segments. "

---

## [Decision Letter · Decision Letter 1]

18 Mar 2020

Impact of law enforcement and increased traffic ticket fines policy on road traffic mortality, injuries and offenses in Iran: Interrupted time series analysis

PONE-D-19-32920R1

Dear Dr. Mohammadzadeh Moghaddam,

We are pleased to inform you that your manuscript has been judged scientifically suitable for publication and will be formally accepted for publication once it complies with all outstanding technical requirements.

With kind regards,

Feng Chen

Academic Editor

PLOS ONE

Additional Editor Comments (optional):

Reviewers' comments:

Reviewer's Responses to Questions

**Comments to the Author**

1. If the authors have adequately addressed your comments raised in a previous round of review and you feel that this manuscript is now acceptable for publication, you may indicate that here to bypass the “Comments to the Author” section, enter your conflict of interest statement in the “Confidential to Editor” section, and submit your "Accept" recommendation.

Reviewer #1: (No Response)

Reviewer #2: All comments have been addressed

2. Is the manuscript technically sound, and do the data support the conclusions?

Reviewer #1: (No Response)

Reviewer #2: Yes

3. Has the statistical analysis been performed appropriately and rigorously? 

Reviewer #1: (No Response)

Reviewer #2: Yes

4. Have the authors made all data underlying the findings in their manuscript fully available?

Reviewer #1: (No Response)

Reviewer #2: Yes

5. Is the manuscript presented in an intelligible fashion and written in standard English?

Reviewer #1: (No Response)

Reviewer #2: Yes

6. Review Comments to the Author

Reviewer #1: The authors have addressed most of my comments. Nonetheless, some minor revisions should be made on the formats of references where there are many inconsistencies and typos. For example, the correct format of Zeng et al. (2018, 2019) should be :

Zeng, Q., Wen, H., Huang, H., Pei, X., Wong, S. C., 2018. Incorporating temporal correlation into a multivariate random parameters Tobit model for modeling crash rate by injury severity. Transportmetrica A: transport science, 14(3), 177-191.

Zeng, Q., Guo, Q., Wong, S. C., Wen, H., Huang, H., Pei, X., 2019. Jointly modeling area-level crash rates by severity: a Bayesian multivariate random-parameters spatio-temporal Tobit regression. Transportmetrica A: Transport Science, 15(2), 1867-1884.

Please check and modify the formats of all references carefully.

Reviewer #2: (No Response)

7. PLOS authors have the option to publish the peer review history of their article (what does this mean?). If published, this will include your full peer review and any attached files.

Reviewer #1: No

Reviewer #2: No

---

## [Editor Report · Acceptance letter]

30 Mar 2020

PONE-D-19-32920R1 

Impact of law enforcement and increased traffic fines policy on road traffic fatality, injuries and offenses in Iran: Interrupted time series analysis 

Dear Dr. Mohammadzadeh Moghaddam:

I am pleased to inform you that your manuscript has been deemed suitable for publication in PLOS ONE. Congratulations! Your manuscript is now with our production department. 

With kind regards,

on behalf of

Dr. Feng Chen 

Academic Editor

PLOS ONE